# A Systematic Review of Endoscopic Treatments for Concomitant Malignant Biliary Obstruction and Malignant Gastric Outlet Obstruction and the Outstanding Role of Endoscopic Ultrasound-Guided Therapies

**DOI:** 10.3390/cancers15092585

**Published:** 2023-04-30

**Authors:** Giacomo Emanuele Maria Rizzo, Lucio Carrozza, Dario Quintini, Dario Ligresti, Mario Traina, Ilaria Tarantino

**Affiliations:** 1Endoscopy Unit, Department of Diagnostic and Therapeutic Services, IRCCS-ISMETT Palermo, 90127 Palermo, Italy; 2Ph.D. Program, Department of Surgical, Oncological and Oral Sciences (Di.Chir.On.S.), University of Palermo, 90133 Palermo, Italy; 3Ilaria Tarantino, Chief of Endoscopy Ultrasound Service, Department of Diagnostic and Therapeutic Services, IRCCS-ISMETT Palermo, 90127 Palermo, Italy

**Keywords:** gastric outlet obstruction, biliary obstruction, GOO, MBO, endoscopic ultrasound, endoscopy, EUS-BD, EUS-GEA, gastroenteroanastomosis, gastrojejunostomy, pancreas, LAMS, cancer

## Abstract

**Simple Summary:**

Palliation of gastric outlet obstruction (GOO) and obstructive jaundice due to malignancy is a fundamental step in improving quality of life (QoL) and initiating or restarting systemic oncologic therapy in these patients. Endoscopic retrograde cholangiopancreatography (ERCP) with biliary stenting is still the primary treatment for malignant biliary obstruction (MBO), but it fails more frequently when GOO is concomitant. Poor quality studies are present in the medical literature on the application of endoscopic ultrasound (EUS) guided procedures among patients suffering from double obstruction, and few studies explore outcomes when endoscopy is performed during the same session or within a few days. Our aim was to systematically search the evidence supporting the use of EUS-guided procedures in this scenario in order to evaluate the best strategy for concomitant treatment of MBO and MGOO.

**Abstract:**

Background: The treatments for cancer palliation in patients with concomitant malignant biliary obstruction (MBO) and gastric outlet obstruction (MGOO) are still under investigation due to the lack of evidence available in the medical literature. We performed a systematic search and critical review to investigate efficacy and safety among patients with MBO and MGOO undergoing both endoscopic ultrasound-guided biliary drainage (EUS-BD) and MGOO endoscopic treatment. Methods: A systematic literature search was performed in PubMed, MEDLINE, EMBASE, and the Cochrane Library. EUS-BD included both transduodenal and transgastric techniques. Treatment of MGOO included duodenal stenting or EUS-GEA (gastroenteroanastomosis). Outcomes of interest were technical success, clinical success, and rate of adverse events (AEs) in patients undergoing double treatment in the same session or within one week. Results: 11 studies were included in the systematic review for a total number of 337 patients, 150 of whom had concurrent MBO and MGOO treatment, fulfilling the time criteria. MGOO was treated by duodenal stenting (self-expandable metal stents) in 10 studies, and in one study by EUS-GEA. EUS-BD had a mean technical success of 96.4% (CI 95%, 92.18–98.99) and a mean clinical success of 84.96% (CI 95%, 67.99–96.26). The average frequency of AEs for EUS-BD was 28.73% (CI 95%, 9.12–48.33). Clinical success for duodenal stenting was 90% vs. 100% for EUS-GEA. Conclusions: EUS-BD could become the preferred drainage in the case of double endoscopic treatment of concomitant MBO and MGOO in the near future, with the promising EUS-GEA becoming a valid option for MGOO treatment in these patients.

## 1. Introduction

Malignancies involving the area between the duodenum, major papilla, head of the pancreas, and distal common bile duct (CBD) can lead to malignant biliary obstruction (MBO) and malignant gastric outlet obstruction (MGOO). When clinical presentation of biliary and duodenal obstruction is simultaneous, these tumors are usually at the late stage of the disease, and the only option is palliative treatment, which is essential for the quality of life (QoL) of these patients. Endoscopic retrograde cholangiopancreatography (ERPC) with transpapillary biliary stent placement is the standard treatment for unresectable MBO [1], though EUS-guided biliary drainage (EUS-BD) is becoming more widespread. EUS-BD is reported in patients with difficult or failed ERCP, and MGOO is one of the major reasons for this failure. In addition, endoscopic treatment of MGOO through duodenal metal stent placement or endoscopic ultrasound gastroenteroanastomosis (EUS-GEA) is increasingly being performed because it is less invasive compared with surgical bypass [2,3,4]. Indeed, open surgical bypass (either biliary or gastrojejunostomy) has significant morbidity (25%) and mortality (2.5%) [5]. Although the double stenting (biliary and duodenal) endoscopic procedure is considered the standard, less invasive, treatment for combined MBO and MGOO, EUS-guided procedures will likely become the main palliative treatment strategy for concurrent MBO and MGOO in the next few years.

The aim of our study was to perform a systematic search, including a comprehensive literature search [6] and critical review, in order to investigate outcomes in terms of efficacy and safety among patients with MBO and MGOO undergoing concurrent (or within 7 days) EUS-guided biliary drainage and MGOO endoscopic treatment.

## 2. Materials and Methods

### 2.1. Procedures

#### 2.1.1. Biliary Drainage

Historically, patients with MBO needing drainage were initially treated with surgery, but, nowadays, minimally invasive techniques are available and preferred; therefore, surgical hepaticojejunostomy as a first-line approach is outdated, even when concurrent MGOO is present. The equipment for endoscopic biliary drainage includes both standard retrograde techniques, such as ERCP for transpapillary drainage and more advanced EUS-guided techniques. In general, EUS-guided biliary drainage includes transmural drainage (EUS-guided hepaticogastrostomy [EUS-HGS], choledochoduodenostomy [EUS-CDS], cholecystoduodenostomy/cholecystogastrostomy [EUS-GBD]) and transpapillary stenting introducing a guidewire under EUS-guidance for a transmural approach (rendezvous [EUS-RV] and antegrade stenting for transpapillary stenting) [7,8,9]. The aforementioned techniques regarding EUS-BD have been reported in detail over the years [7,10], and, nowadays, they are standardized. In fact, they are already in clinical practice in tertiary centers all over the world, and they are starting to spread to even more centers. In general, it is fundamental to have extensive experience in these procedures in order to understand the main issues: indications; the best approach, if transgastric or transduodenal; scope stability; safe ultrasonographic window; correct identification of the desired biliary target (common bile duct, intrahepatic ducts, or gallbladder, depending if CDS, HGS, or GBD, respectively); proper devices (e.g., guidewires, needle, stent, etc.) and skills in complication management [10,11,12,13].

#### 2.1.2. Treatment of MGOO

Considering the anatomical location of the duodenal stricture in relation to the papilla, GOO and biliary strictures are classified into the three types of bilio-duodenal strictures [14]. Treatment of the concurrent double obstruction could be dependent on the type of double stricture according to this classification (Table 1). Regarding the treatment of MGOO, surgical gastrojejunostomy bypass has been the most common option in the past [5,15], but, nowadays, endoscopic options are available and are effective thanks to consolidated endoscopic enteral stenting [16] and the development of novel and more advanced techniques, such as NOTES (natural orifice transluminal endoscopic surgery) and EUS-guided gastroenteroanastomosis (EUS-GEA). Endoscopic stenting has been useful in treating MGOO thus far, but it has a high rate of reintervention due to the low time of patency compared to gastrojejunostomy [17]. On the other hand, NOTES is still under development; it has been proven to be effective, mostly in porcine models [18,19], yet it is an option as a rescue therapy in the case of AEs during EUS-GEA. EUS-GEA has been optimized since 2011, when the lumen apposing metal stent (LAMS), a fully-covered short metal stent with double flanges capable of joining two lumen, creating a direct luminal connection, was developed [20]. Conceptually, EUS-GEA is performed by, first, advancing a catheter (or a double balloon/single balloon enteric tube) over a stiff guidewire through the duodenal stricture. Then saline is injected through the catheter downstream of the stricture (single balloon technique) or between the inflated balloons (double balloon technique) in order to fill the target jejunal lumen. Finally, after EUS-identification of the enlarged enteral loop (“target”), the distal flange of the LAMS is deployed into the jejunal lumen (using the hands-free technique or through previous placement of a guidewire after loop puncture), and the proximal flange is deployed into the gastric lumen (with or without the intra-channel release technique) [21,22,23]. This EUS-guided technique showed a technical success rate of 90%, confirming the feasibility of this application of EUS-GEA, even though it is still dependent on the expertise of skilled endosonographers [22,24,25].

### 2.2. Data Source and Literature Searches

This work is reported according to the Preferred Reporting Items for Systematic Reviews and Meta-Analyses (PRISMA) Statement [26], and the search strategy included the use of the four-part PICO model [27]. Rayyan was used to identify eligible studies and for the screening process [28]. A systematic literature search was then performed by three reviewers (G.E.M.R., D.Q., and L.C.) in PubMed (MEDLINE), EMBASE, and the Cochrane Library through the use of strings including “biliary obstruction”, “gastric outlet obstruction”, “malignant”, “EUS”, “endoscopic ultrasound”, “endoscopy”, “endoscopic stenting”, “gastrojejunostomy”, “gastroenteroanastomosis”, and “biliary drainage” (more details are in Appendix A). The literature search included studies published until February 2023 and, moreover, the computer search was supplemented with manual searches of the reference lists of the reviews and studies retrieved, in order to identify additional studies. When the results of the same cohort were analyzed in more than one publication, only the most recent and complete data were included in the review. Finally, a cross-reference check from the retrieved studies was performed to identify duplicated reports.

### 2.3. Eligibility Criteria

Studies were considered eligible in the systematic review if they met the following criteria: (1) they included patients with malignancy of any etiology (e.g., pancreatic, biliary, duodenal, gastric, and so on) developing biliary obstruction and gastric outlet obstruction (GOO); (2) they included endoscopic biliary treatments, such as EUS-guided biliary drainage; (3) they included the following MGOO treatments: endoscopic enteral stenting, EUS-guided gastroenteroanastomosis, or surgical gastroenteroanastomosis; (4) they included patients with concomitant biliary drainage and treatment of MGOO, defined as being performed during the same hospital stay, considering a time interval of no more than seven days between procedures; (5) they evaluated efficacy outcomes in terms of technical and/or clinical success and/or safety. Studies were excluded if: (1) they included patients with recurrence of MGOO (failure of first MGOO treatment); (2) they had overlapping data; (3) they were case reports, case series (<5 patients), or review articles; (4) they were abstracts or posters at international meetings; (5) they were not written in English; or (6) if they included patients with altered anatomy of the upper GI tract due to surgery before developing malignancy. Both experimental and observational studies (either prospective or retrospective) without respect to their primary objectives were included.

### 2.4. Study Selection and Data Collection Process

After the removal of duplicates, data extraction included study–and patient-level variables, such as study design, geographical area, number of centers, number of total patients, number of patients undergoing concomitant (or within 7 days) biliary and duodenal treatment, type of malignancy, type of biliary and MGOO intervention, technical and clinical success, survival, AEs, recurrent biliary obstruction, recurrent duodenal obstruction, and reinterventions. Studies reporting “temporary” stenting were excluded.

### 2.5. Outcome Measures

Technical success is defined as adequate placement of stents in biliary and duodenal procedures. Clinical success of biliary stenting is defined as a reduction in serum bilirubin level within two weeks post-drainage. Clinical success of treatment for MGOO is mainly referred to as an improvement in the score on the gastric outlet obstruction scoring system (GOOSS) after treatment or, when GOOSS was not applied, the improvement in the quality of oral food intake. Technical and clinical success were determined for biliary stenting/bypass and duodenal stenting/bypass both together and separately, when possible. Procedure-related AEs were various and heterogeneous, and data extraction was based on the AE terms indicated in each study. When available, data on recurrent biliary obstruction (RBO), defined as a composite endpoint including both stent patency and migration or dysfunction according to the Tokyo criteria [29], and recurrent duodenal obstruction (RDO), defined as reoccurrence of GOO symptoms, were extracted. The AE rate was given as the number of patients with one or more adverse events out of the total number of patients fulfilling the inclusion criteria. Reinterventions, when available, were investigated and defined as the number of patients who required endoscopic or surgical intervention after RBO or RDO.

## 3. Results

### 3.1. Literature Search Results

Our primary search identified 654 articles overall. We excluded 323 studies because they were not consistent with our aim, including reviews, case reports, case series, editorials, and letters to the editor. In addition, duplicate articles (n = 168) were removed. After the identification and screening process, 54 of the initial studies were reviewed for inclusion and exclusion criteria (Figure 1). Finally, 11 studies were selected for systematic review.

### 3.2. Study Characteristics

Table 2 shows the characteristics of the 11 studies [30,31,32,33,34,35,36,37,38,39,40] included in the systematic review. Overall, 337 patients with MBO and GOO were identified. The number of patients included in single studies varied, ranging from 5 to 110. The number of patients with concurrent MBO and MGOO following inclusion and exclusion criteria extracted from the articles was 150 (44.5% of total patients).

### 3.3. Study-Level Variables

Five studies were performed in Asian countries (55.6%) [30,33,34,35,38] and six in Western countries [31,32,36,37,39,40]. Most of the studies (n = 8, 72.7%) were single-center, and three were multicenter [36,39,40]. Among EUS-BD procedures, the most common was EUS-CDS. EUS-HGS was performed in five studies [34,35,36,37,39]. MGOO was treated with duodenal endoscopic stenting in ten studies (90.9%), while EUS-GEA was performed in only one (Appendix A) [39]. In general, no significant difference in the definition of technical success was seen among studies, both considering biliary and duodenal procedures (Appendix A). Procedures were performed simultaneously in seven studies (63.6%), while, in four studies, they were performed within seven days of each other [34,36,37,40].

### 3.4. Patient-Level Variables

Patients with pancreatic cancer were most commonly found among the studies, followed by biliary and gastrointestinal tumors (Table 2). Patients with gallbladder tumors were included in two studies [36,40], but, in one of these [36], the patients did not present with simultaneous biliary and duodenal obstruction. In some studies, metastatic tumors from different sites were present as well (cervix squamous cell cancer, colonic adenocarcinoma, breast cancer, and others) [31,34,36,40]. In patients undergoing EUS-BD, the type of stent varied greatly among studies (Appendix A).

### 3.5. Outcomes

#### 3.5.1. Technical and Clinical Success

EUS-BD showed a mean technical success of 96.4% (CI 95%, 92.18–98.99) and a mean clinical success of 84.96% (CI 95%, 67.99–96.26). Duodenal stenting was technically successful in all ten of the studies where it was used. Technical success was 95.6% for EUS-GEA with 100% clinical success (vs 87.95% [CI 95%, 67.7–98.92] for duodenal stenting). No data regarding RDO and RBO were extractable regarding the articles included in our review, so reintervention was not extractable either.

#### 3.5.2. Safety

AEs of EUS-BD procedures had a mean occurrence of 28.73% (CI 95%, 9.12–48.33) among the studies in which data were reported. The type of AEs varied from abdominal pain and fever to cholangitis, bile leakage, bleeding, cardiogenic shock, and pneumoperitoneum (Appendix A), and, furthermore, all of them were reported as BD-related AEs (no AEs were reported regarding treatment of MGOO).

## 4. Discussion

Malignant biliary obstruction and gastroenteral obstruction are two of the most frequent complications associated with primary biliary and pancreatic tumors and metastatic disease [36,39,40,41] leading to the development of jaundice and MGOO. Palliation of MGOO and obstructive jaundice is a fundamental step in improving quality of life (QoL) and initiating or restarting systemic oncologic therapy in these patients. ERCP with biliary stenting is still the first-line treatment for MBO in the real world, but it may fail more commonly when MGOO is concomitant. EUS-BD is considered a feasible alternative after ERCP failure or when a transpapillary approach is not possible [42]; furthermore, its efficacy as a primary treatment of MBO has already been highlighted [43,44,45], and it is under investigation in ongoing randomized trials [NCT04099862, NCT03870386, NCT04898777]. Despite this evidence and the presence of other BD-related issues (duodenobiliary reflux [46] and subsequent cholangitis or stent occlusion [47]), EUS-BD is still subordinate to local expertise and preference. As for MGOO, endoscopic stenting and surgical GJ demonstrated comparable rates of technical success, clinical success, and reintervention, but patients undergoing surgical GJ had longer survival and luminal patency [17].

Unfortunately, most of the evidence in the literature concerning concurrent bilioenteral obstruction is of poor quality and does not include the use of EUS-guided procedures; therefore, this setting is still a grey area, and the best therapeutic strategy in this scenario has yet to be deeply analyzed. To our knowledge, no reports with a systematic search and critical review focusing on EUS-BD in patients with concurrent MBO and MGOO are present in the literature thus far. The presence of MGOO limits some therapeutic biliary options and may affect outcomes. Our systematic review identified 11 studies, including 150 patients suffering from concurrent MBO and MGOO. In our qualitative analysis, we focus on highlighting the advantages of performing both procedures (biliary and enteral) during the same session or, if different sessions were needed, during the same hospital stay (maximum time between procedures was conventionally set at 7 days). Double treatment can be technically difficult depending on the location of MBO and MGOO [14], so classification of the double stricture is helpful to identify the best therapeutic options (Table 1). Furthermore, it is mandatory to focus on the advantages and disadvantages of each procedure in order to select the best treatment for each patient. In fact, a therapeutic algorithm based on the classification of obstructions has previously been proposed [48], even though this algorithm considered only duodenal stenting for treating GOO. In the present work, the included studies were mostly single-center (n = 8). There were only 3 multicenter studies, and had the peculiarity of being performed after 2018 (one in 2018, one in 2021, and another in 2022) [36,39,40], all showing a positive trend towards collaboration in identifying the correct strategy in this particular scenario. Another retrospective international multicenter study was recently published [49] analyzing different strategies for treating both obstructions in 93 patients undergoing double endoscopic treatment (median interval between procedures: 41 [IQR 5–68] days), demonstrating the need for further exploration into well-designed prospective clinical studies.

Regarding MGOO, among those patients with double obstruction, we found that endoscopic enteral stenting was the treatment of choice in most of the studies (n = 10), with only one study reporting the use of EUS-GEA, even though it shows high rates of technical (95.6%) and clinical success (100%) [39]. The low rate of EUS-GEA among studies could be explained by the lack of diffusion of this advanced technique, which requires expertise and a skilled endosonographer. Nonetheless, we found that duodenal stenting was technically successful (100% technical success shown in 8 studies) with a clinical success rate of 87.95% (CI 95%, 67.7–98.92) among those studies in which data were extractable for this measure (n = 4), a limitation of our search. In general, the advantages of EUS-guided GEA are its minimal invasiveness, safety (Canakis et al. reported no AEs after EUS-GEA among 21 patients with MGOO), and high technical and clinical success rates (95.6% and 100%, respectively) [39]. Considering this evidence and that reported by Chen et al. [50], it is fair to consider EUS-GEA, in the near future, as the preferred procedure for GJ in patients with MGOO. Moreover, growing evidence supports a better efficacy of EUS-GEA over enteral stents in terms of clinical success and symptom recurrence in this set of patients, as in a recent multicenter propensity score-matched comparison, which showed better outcomes in EUS-GEA vs. duodenal stenting (technical success rates: 94% (CI 95%, 89–99) vs. 98% (CI 95%, 95–100; *p* = 0.44); clinical success rates: 91% (CI 95%, 85–97) vs. 75% (CI 95%, 66–84 *p* = 0.008); and stent dysfunction: 1% (CI 95%, 0–4) vs. 26% (CI 95%, 15–37; *p* < 0.001) [51]).

As regards EUS-BD, we found that EUS-CDS was performed mostly by placing self-expandable metal stents (SEMS) rather than LAMS (plastic biliary stents were even placed in one study [30]), probably, as mentioned above for EUS-GEA, due to lack of expertise and difficulty of device acquisition. Nowadays, LAMS is the preferred option in the case of EUS-CDS [52,53], and an international multicenter study from Mangiavillano et al. [40] even demonstrated the technical feasibility of the placement of a LAMS through the mesh (TTM) of duodenal stents, with only one failure among the 23 patients included (4.3%) and no AEs reported. It should be noted that, in the latter study, EUS-CGD (choledocogallbladder drainage) was performed TTM in 14 patients (60.9%) with 100% technical success and no AEs reported [40]. On the other hand, EUS-HGS was used in six studies (54.5%) [34,35,36,37,38,39], confirming that an EUS-guided transgastric approach for biliary drainage is one of the preferred options in patients with duodenal obstruction. Although EUS-HGS is likely to have better stent patency, its safety is still under debate due to the high AE rate reported so far, though an increase in expertise could reduce this number. While further prospective studies should be mandatory in the coming years, patients with double obstruction are strong candidates for EUS-HGS [54]. Ogura et al. [34] compared EUS-CDS and HGS in this scenario, showing an increased risk of duodenobiliary reflux when performing EUS-CDS due to the close location of duodenal stent and bilio-duodenal fistula. Considering safety, EUS-guided transmural BD is safer than a transpapillary approach as regards post-procedure pancreatitis, so it should be considered especially when approaching patients without pancreatic duct obstruction. On the other hand, when EUS-CDS is performed, and the duodenal stent is on-site, the presence of the latter is significantly associated with a higher risk of biliary reobstruction during the follow-up (odds ratio [OR]: 3.6, CI 95%, 1.2–10.2; *p* = 0.018), as reported in a recent prospective study [53]. Moreover, our included studies reported further complications, such as abdominal pain, cholangitis, bile leakage, cardiogenic shock, and pneumoperitoneum. In general, the percentage of EUS-BD related AEs varied greatly, from 0% [32,40] to 60% [31], while no AEs regarding the treatment of MGOO were reported (Appendix A). The duodenal stent, which is more standardized and commonplace, was used more than an EUS-guided approach, so this could explain the differences between the rates of BD-related, only EUS-guided, and MGOO treatment-related AEs. In any case, cholangitis was the most common AE reported, bleeding was reported in only one case concurrently with cholangitis [36], and biliary stent dislodgment was reported in three cases [36,39].

However, the limitations of our study included the lack of individual data regarding technical and clinical success in most of the studies and the heterogeneity of patients in terms of differences in cancer typologies, which could confound aggregate data regarding outcomes, which is another limitation of the studies included. Reporting bias is present due to missing extractable information, such as RDO and RBO. In addition, we used the PICO model as a powerful tool to have a methodologically correct search strategy [27], but since its efficacy has recently been reconsidered in searching databases such as Embase and PubMed, especially in the field of upper GI and pancreatic diseases [55], this could be a limitation of our study.

## 5. Conclusions

In conclusion, EUS-guided procedures, EUS-BD and EUS-GEA, are likely to spread quickly in clinical practice for the treatment of concurrent MGOO and MBO because their major advantage is that stents can be placed away from the biliary and duodenal obstructions. Recurrence of both biliary and enteral obstruction could be lower in the EUS-guided approach rather than in transpapillary biliary stents or duodenal stents due to tumor ingrowth or overgrowth, but further prospective studies are needed to confirm these outcomes, especially among patients undergoing both procedures in the same session. On the other hand, EUS-guided procedures have an increased stent migration rate because the stents are placed between two adjacent but non-attached organs, and AEs such as stent dislocation can be serious, leading to perforation or peritonitis. The EUS-GEA procedures reported in the literature have only been performed by experts, as it is a relatively novel technique, so its generalizability needs to be clarified, and the development of dedicated devices could help the standardization of this useful and minimally-invasive procedure. In summary, EUS-BD could potentially be the standard procedure for combined MBO and GOO in terms of both technical factors and clinical outcomes [38,39,56]. Furthermore, we believe that, as evidence on EUS-GEA increases, EUS-guided double stenting is likely to become the standard procedure for combined MBO and GOO in the near future.

## Figures and Tables

**Figure 1 cancers-15-02585-f001:**
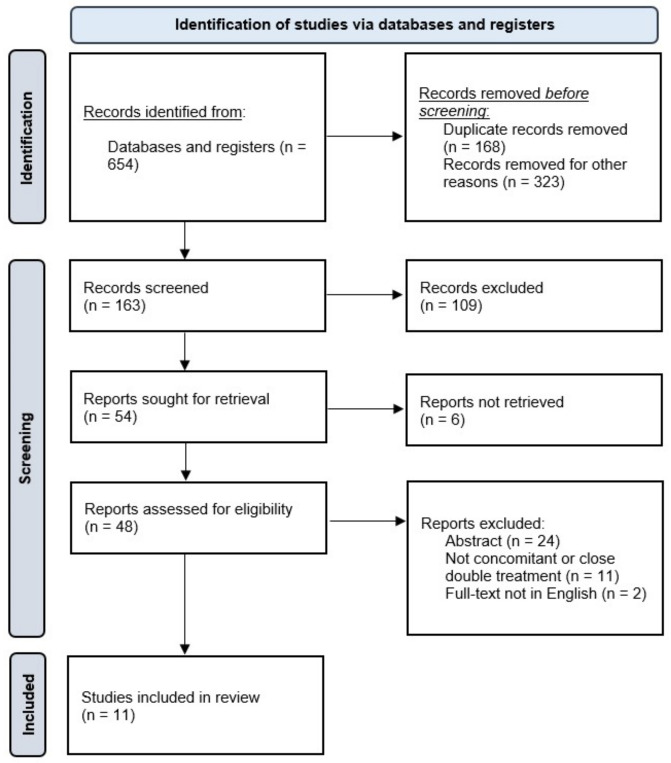
Flowchart of search and screening process: 11 studies were included in the review.

**Table 1 cancers-15-02585-t001:** Classification based on location of malignant stricture in concurrent malignant biliary obstruction (MBO) and gastric outlet obstruction (MGOO) and treatment suggestions. Modified from [14].

Treatment Suggestion	Description	Classification
♦ Transpapillary stenting (ERCP) is possible if the scope can pass the duodenal obstruction or the duodenal stent ♦ EUS-guided antegrade (AG) stent placement is preferred over EUS-guided transmural stenting ♦ Duodenal stenting/EUS-GEA	GOO occurs at the level of the *duodenal bulb* or *upper duodenal genu*, but *without* involvement of the papilla	**Type 1**
♦ Double stenting is technically possible by adding a duodenal stent if an indwelling biliary stent is present ♦ In cases with the ampulla overlapped by a duodenal stent, there are some techniques for biliary access: ○ In cases with an indwelling duodenal stent, biliary access can be achieved by the rendezvous technique, either EUS or PTBD-guidance ○ Simultaneous double stenting with the temporary plastic biliary stent placement technique ♦ Placement of a duodenal stent followed by a transpapillary biliary stent through the duodenal stent (to enable future reinterventions for biliary stent occlusion) ♦ EUS-CDS and EUS-GEA ♦ EUS-HGS and duodenal stenting/EUS-GEA	GOO affects the *second part of the duodenum*, *with* involvement of the papilla.	**Type 2**
♦ Transpapillary stenting including both ERCP and EUS-guided antegrade stenting is possible but prone to duodenobiliary reflux ♦ EUS-CDS and duodenal stenting/EUS-GEA ♦ EUS-HGS and duodenal stenting/EUS-GEA	GOO involves the third part of the duodenum, distal to and without involvement of the papilla.	**Type 3**

MBO = malignant biliary obstruction; ERCP = endoscopic retrograde cholangiopancreatography; MGOO = malignant gastric outlet obstruction; EUS-CDS = endoscopic ultrasound choledocoduodenostomy; PTBD = percutaneous transhepatic biliary drainage; EUS-HGS = endoscopic ultrasound hepaticogastrostomy.

**Table 2 cancers-15-02585-t002:** Characteristics of the included studies (n = 11).

Study, Year	Design and Centers	Geographical Area	No. of Patients	No. of Patients Reflecting Our EC and IC	Type of Malignancy	Type of BD	Type of Biliary Stent	Treatment of MGOO	First Procedure Performed	Time between the Two Procedures	Follow-Up	AEs, n (%)
**Iwamuro,** **2010** [30]	Retrospective/ Single-center	Eastern	7	2	Pancreatic (n = 2)	EUS-BD	Plastic stent	Duodenal stenting	MGOO	Simultaneous	46.4 w and 9.9 w	1 (50)
**Maluf-Filho,** **2012** [31]	Retrospective/ Single-center	Western	5	5	Pancreatic (n = 3) Cervix squamous cell cancer (n = 1) Colonic adenocarcinoma (n = 1)	EUS-BD	PC-SEMS (n = 4) and U-SEMS (n = 1)	Duodenal stenting	Biliary	Simultaneous	3 m + 17 d + 2 m + 4 d + 15 d	3 (60)
**Rebello,** **2012** [32]	Prospective/ Single-center	Western	7	7	Pancreatic (n = 7)	EUS-BD	PC-SEMS	Duodenal stenting	Biliary	Simultaneous	140 days	0 (0)
**Tonozuka,** **2013** [33]	Retrospective/ Single-center	Eastern	11	4	Pancreatic (n = 4)	EUS-BD	C-SEMS	Duodenal stenting	MGOO	Simultaneous	Survival time: 37 d, 74 d, 23 d, 69 d	2 (50)
**Ogura,** **2016** [34]	Retrospective/ Single-center	Eastern	39	39	EUS-CDS group: 11 pancreaticobiliary cancers and 2 others; EUS-HGS group: 21 pancreaticobiliary cancers and 5 others	EUS-CDS and EUS-HGS	FC metal stent	Duodenal stenting	MGOO	Within 7 days	OS: EUS-CDS median 98 days, EUS-HGS median 133 days	8 (20.5)
**Sato,** **2016** [35]	Retrospective/ Single-center	Eastern	43	17	NA	EUS-CDS (n = 16) EUS-HGS (n = 1)	C-SEMS	Duodenal stenting	NA	Simultaneous	Death or 90 days	NA
**Hamada,** **2018** [36]	Retrospective/ Multicenter	Western	110	20	Pancreatic (n = 72) Biliary (n = 9) Gastric (n = 9) Ampullary (n = 9) Gallbladder (n = 9) Others (n = 9)	EUS-CDS (n = 10) EUS-HGS (n = 10)	SEMS, Plastic	Duodenal stenting	NA	Simultaneous or within 7 days	450 days, median	7 (35%)
**Debourdeau,** **2021** [37]	Retrospective/ Single-center	Western	31	7	NA	EUS-HGS (n = 11) EUS-CDS (n = 1)	PC-SEMS	Duodenal stenting	Biliary Or MGOO	Simultaneous or within 7 days	NA	NA
**Mangiavillano,** **2021** [40]	Retrospective/ Multicenter	Western	23	23	Pancreatic adenocarcinoma (n = 13) Advanced ampulloma (n = 2) Metastasis (n = 2) Biliary tumor (n = 2) Pancreatic NET (n = 1) Duodenal adenocarcinoma (n = 1) Gallbladder neoplasia (n = 1) Recurrence of a previous distal esophageal adenocarcinoma (n = 1)	EUS-CDS (n = 9) EUS-GDS (n = 14)	LAMS	Duodenal stenting	Duodenal stenting	Simultaneous or within 7 days	241 days, median	0 (0)
**Canakis,** **2022** [39]	Retrospective/ Mulitcenter	Western	23	21	Pancreatic cancer (n = 16) Breast metastasis (n = 2) Cholangiocarcinoma (n = 2) Colon (n = 1)	EUS-HGS	FC-SEMS	EUS-GEA	BiliaryOrMGOO	Simultaneous	78 days, median	3 (14.3)
**Sasaki,** **2022** [38]	Retrospective/ Single-center	Eastern	38	5	Pancreatic (n = 28)	EUS-CDS (n = 1) EUS-HGS (n = 4)	PC-SEMS	Duodenal stenting	NA	Simultaneous	NA	NA

EC = exclusion criteria; IC = inclusion criteria; BD = biliary drainage; EUS-BD = endoscopic ultrasound biliary drainage; PC-SEMS = partially covered self-expandable metal stent; MGOO = malignant gastric outlet obstruction; AEs = adverse events; C-SEMS = covered self-expandable metal stent; FCSEMS = fully covered self-expandable metal stent; NA = Not Available; LAMS = lumen apposing metal stent; EUS-CDS = endoscopic ultrasound choledocoduodenostomy; EUS-GDS = endoscopic ultrasound gallbladderduodenostomy; EUS-HGS = endoscopic ultrasound hepaticogastrostomy; OS = overall survival.

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
