# Peer review of "A Systematic Review of Endoscopic Treatments for Concomitant Malignant Biliary Obstruction and Malignant Gastric Outlet Obstruction and the Outstanding Role of Endoscopic Ultrasound-Guided Therapies"

_cancers, 2023, doi:10.3390/cancers15092585_

Round 1

Reviewer 1 Report

This title should be more appropriate:

A Review on Endoscopic Treatments for concomitant Malignant Biliary Obstruction and Malignant Gastric Outlet Obstruction and the Outstanding Role of EUS-guided Therapies.

Instead of  mGOO writte MGOO

Try to describe the procedures more clearly and avoid repetition. The manuscript could also be summarized and reduced

English language must be corrected by a native speaker o person with extensive knowledge 

Author Response

This title should be more appropriate:

A Review on Endoscopic Treatments for concomitant Malignant Biliary Obstruction and Malignant Gastric Outlet Obstruction and the Outstanding Role of EUS-guided Therapies.

Answer: Thank you for your suggestion, which we appreciated, so we modified the title.

Instead of mGOO write MGOO

Answer: We changed it.

Try to describe the procedures more clearly and avoid repetition. The manuscript could also be summarized and reduced

Answer: Thank you for your comments. We revised the manuscript trying to be briefer according to your suggestions. We added more details on the technical aspects of the procedures.

Comments on the Quality of English Language

English language must be corrected by a native speaker o person with extensive knowledge

Answer: Our English language service revised the manuscript.

Reviewer 2 Report

This manuscript describes the content of endoscopic ultrasound-guided treatment for malignant biliary obstruction and gastric outlet obstruction, and I consider it interesting. I have some comments to the author.

Comment 1. Although this manuscript is submitted as a “Review”, it is a systematic review, and I think it would be better to submit the PRISMA_2020_checklist as supplementary material.

Comment 2. Abbreviations in Table 1 should be explained in foot not.

Comment 3. The results state that “EUS-BD had a mean technical success of 99.1% (CI 95%, 97.8-100.4) and a mean clinical success of 86.3% (CI 95%, 68.5-104.1),” but % , isn't it strange that the 95% CI upper limit exceeds 100%?

Author Response

This manuscript describes the content of endoscopic ultrasound-guided treatment for malignant biliary obstruction and gastric outlet obstruction, and I consider it interesting. I have some comments to the author.

Comment 1. Although this manuscript is submitted as a “Review”, it is a systematic review, and I think it would be better to submit the PRISMA_2020_checklist as supplementary material.

Answer: Thank you for your accurate revision. As you said, actually, we performed a literature search following PRISMA statements, so we added the checklist to the supplementary materials.

Comment 2. Abbreviations in Table 1 should be explained in foot not.

Answer: We added them.

Comment 3. The results state that “EUS-BD had a mean technical success of 99.1% (CI 95%, 97.8-100.4) and a mean clinical success of 86.3% (CI 95%, 68.5-104.1),” but % , isn't it strange that the 95% CI upper limit exceeds 100%?

Answer: Thank you for your revision. We apologize for it, because it was a mistake occurring while copying the correct paragraph into the template for submission. However, we revised the analysis and corrected the text as follow “EUS-BD showed a pooled technical success of 96.4 (CI 95%, 92.18 to 98.99) and a pooled clinical success of 84.96% (CI 95%, 67.99-96.26).”

Reviewer 3 Report

I read with interest the article from Rizzo at al, focusing on EUS-guided treatment of concomitant gastric outlet obstruction and malignant biliary stricture.

It is a well written review, with rigorous literature search and analysis. In discussion, the main issues related to the use of novel UES-guided techniques are adequately addressed.

I have few comments:

Abstract: SEMS is for ‘self-expanding metal stents’, please correct.

Discussion:

First sentence: Malignant biliary obstruction and gastro-enteral obstruction are associated not only to primary biliary and pancreatic cancers, but also to metastatic disease. In fact, many patients of the series included in the review are affected from metastatic cancer to the pancreatoduodenal area. The reference [40] only refers to pancreatic cancer. Please rewrite this paragraph.

I think that the most common complication related to the double stenting, such as cholangitis, stent migration, pancreatitis, and bleeding should be discussed.

Last paragraph: Another review on the same topic has been published in 2020 (doi.org/10.5946/ce.2019.050). That article should be mentioned and included in discussion.

Author Response

I read with interest the article from Rizzo at al, focusing on EUS-guided treatment of concomitant gastric outlet obstruction and malignant biliary stricture.

It is a well written review, with rigorous literature search and analysis. In discussion, the main issues related to the use of novel UES-guided techniques are adequately addressed.

Answer: Thank you for your kind appreciation. We followed your suggestions, which we strongly consider useful and appropriate.

I have few comments:

Abstract: SEMS is for ‘self-expanding metal stents’, please correct.

Answer: We changed it.

Discussion: First sentence: Malignant biliary obstruction and gastro-enteral obstruction are associated not only to primary biliary and pancreatic cancers, but also to metastatic disease. In fact, many patients of the series included in the review are affected from metastatic cancer to the pancreatoduodenal area. The reference [40] only refers to pancreatic cancer. Please rewrite this paragraph.

Answer: We corrected the sentence.

I think that the most common complication related to the double stenting, such as cholangitis, stent migration, pancreatitis, and bleeding should be discussed.

Answer: We improved the discussion focusing on your suggestions.

Last paragraph: Another review on the same topic has been published in 2020 (doi.org/10.5946/ce.2019.050). That article should be mentioned and included in discussion.

Answer: Thank you for the suggestion. We mentioned it in the discussion.

Round 2

Reviewer 1 Report

Thank you for improving your manuscript